# Free-space creation of ultralong anti-diffracting beam with multiple energy oscillations adjusted using optical pen

Xiaoyu Weng[1,2], Qiang Song [3], Xiaoming Li[1], Xiumin Gao[2], Hanming Guo[2], Junle Qu[1] & Songlin Zhuang[2]

A light beam propagating over an infinite anti-diffracting distance requires infinite power to preserve its shape. However, the fundamental barrier of finite power in free space has made the problem of diffraction insurmountable in recent decades. To overcome this limitation, we report an approach that employs the multiple energy oscillation mechanism, thereby permitting the creation of a light beam with an ultralong anti-diffracting distance in free space. A versatile optical pen is therefore developed to manipulate the number, amplitude, position and phase of energy oscillations for a focusing lens so that multiple energy oscillations can be realized. A light beam with a tunable number of energy oscillations is eventually generated in free space and propagates along a wavy trajectory. This work will enable the extension of non-diffractive light beams to an expanded realm and facilitate extensive developments in optics and other research fields, such as electronics and acoustics.

[1] Key Laboratory of Optoelectronic Devices and Systems of Ministry of Education and Guangdong Province, College of Optoelectronic Engineering, Shenzhen University, Shenzhen 518060, China. [2] Engineering Research Center of Optical Instrument and System, Ministry of Education, Shanghai Key Lab of Modern Optical System, School of Optical-Electrical and Computer Engineering, University of Shanghai for Science and Technology, 516 Jungong Road, Shanghai 200093, China. [3] Optics Department, Telecom Bretagne, Institut Mines-Telecom, IMT Atlantique, CS83818 29238 Brest Cedex 03, France. Correspondence and requests for materials should be addressed to X.W. (email: xiaoyu@szu.edu.cn) or to H.G. (email: hmguo@usst.edu.cn) or to J.Q. (email: jlqu@szu.edu.cn)

   

Diffraction is a natural property of light beams that tends to broaden the beams during propagation in free space, and fighting against the diffraction effect is one of the most important topics in the community of optics. However, it was not until 1987 that the advent of Bessel beam brings the hope of addressing successfully this issue[1], and great success has been achieved in the study of anti-diffracting light beams since then[1–13]. Every breakthrough regarding these light beams leads to the development of many applications, including optical imaging[14–16], optical trapping[17–19], optical communication[20–22], and laser-assisted guiding of electric discharges[23]. All advances are associated with the long anti-diffracting distances of these light beams. For many years, the scientific consensus has been that ideal diffraction-free light beams require infinite power to maintain their shapes during propagation in free space[6,7,11]. Nevertheless, the finite power in free space inhibits the creation of light beams with ultralong anti-diffracting (UAD) distances (see Supplementary Note 1). Although great efforts have been made to generate light beams with long non-diffractive distances, such as Optical needle[11–13], Bessel beam[1,2], and Airy beam[6–9], all share very limited anti-diffracting distances in free space. Among them, Airy beams possess the longest anti-diffracting distances, though they are no longer than $32\lambda$ in free space for a numerical aperture (NA) of 0.8 (see Supplementary Figure 13).

The optical nonlinearity induced by the interaction between light beams and special materials can be used as an alternative to suppress the diffraction effect so that light beams can preserve their shapes without divergence over a long distance[24–29]. For example, light beams that propagate over more than 1000 Rayleigh lengths can be achieved due to the nonlinear response of dipolar glass[29]. A spatial light soliton can also propagate invariantly in other nonlinear materials, where the diffraction effect can be compensated by the self-focusing effect[24–28]. Nevertheless, since such light beams can exist only inside the nonlinear material in which the optical nonlinearity occurs, their utilization in many applications is limited. For example, light beams in nonlinear material cannot excite the fluorescent material inside a cell during the imaging process[14,15]. Although air has been demonstrated to represent a valid nonlinear material to support light bullets, the light bullets' anti-diffracting distances are still restricted by finite apertures, namely, finite power in free space[30,31]. Undoubtedly, the generation of light beams with UAD distances in free space, not just in a particular material, is of great value but still far beyond our understanding.

Here, we report an approach that overcomes this fundamental barrier by employing a multiple energy oscillation mechanism to generate light beams with UAD distances in free space. Based on the energy oscillation mechanism, the main reason of the finite anti-diffracting distance is that when an anti-diffracting light beam completely discharges its energy, it cannot recharge again. A versatile optical pen is therefore developed to manipulate the number, amplitude, position, and phase of energy oscillation so that energy recharge can occur in free space and multiple energy oscillations can be realized. Eventually, UAD light beams with a tunable number of energy oscillations can be obtained, which to a large extent avoids the restriction of finite power in free space.

## Results

**Energy oscillation of anti-diffracting light beams**. An ideal anti-diffracting light beam can propagate without divergence limitlessly in free space; thus, infinite power is necessary to preserve its shape. However, being restricted by the finite power in free space, until now, only a light beam with a finite non-diffractive distance has been obtained in practice[6–9]. It is debatable whether a light beam with a UAD distance can be created in free space. The key to this problem is the new physical insight regarding the propagation of anti-diffracting light beam in free space.

In theory, anti-diffracting light beams can be considered the Fourier transformation of their corresponding Fourier spectra[1,4,6,7]. For example, cubic phase and high-pass pupil filters are the Fourier spectra of the Airy beam[6,7] and Bessel beam[1], respectively. The process of Fourier transformation can be flexibly achieved by an objective lens[32]. Thus, all anti-diffracting light beams can be created in the focusing system of Supplementary Figure 2. According to the general focusing theory in Supplementary Equation 3, one can readily obtain the energy fluxes of anti-diffracting light beams at two symmetrical points $P_1(x, y, z)$ and $P_2(x, y, -z)$, which can be expressed as (see Supplementary Equation 8)

$$\langle \mathbf{S} \rangle_{tP_1} = -\langle \mathbf{S} \rangle_{tP_2}. \tag{1}$$

Thus, when propagating in free space, the light beams experience two inverse energy processes that transfer the energy from $\langle \mathbf{S} \rangle_{tP_2}$ to $\langle \mathbf{S} \rangle_{tP_1}$. Here, $\langle \mathbf{S} \rangle_{tP_2}$ and $\langle \mathbf{S} \rangle_{tP_1}$ are referred to as energy charge and discharge, respectively.

Energy charge and discharge compose an entire energy oscillation, which is a directional energy flux that confines the energy into an interaction between mainlobe and sidelobes. Thus, the light beams would not diverge freely like that of Gaussian beams in free space. For example, a quasi-Airy beam exhibits one energy oscillation from the mainlobe to sidelobes and then from the sidelobes to mainlobe in Supplementary Figure 4, whereas the Bessel beam inverses in Supplementary Figure 7. If only energy oscillation occurs, all anti-diffracting light beams can preserve their shapes without divergence. Even when encountering an obstacle, the mainlobe can carry out self-healing with the power from the sidelobes[8]. This is the reason why anti-diffracting light beams are naturally composed of a mainlobe and sidelobes. Moreover, since all those light beams can be created by focusing their corresponding Fourier spectra in Supplementary Figure 2, energy oscillation is therefore a general property for all anti-diffracting light beams.

**Conceptual change via the energy oscillation mechanism**. Based on the energy oscillation shown in Eq. (1), an anti-diffracting light beam can be divided into two symmetrical parts according to whether the energy flux is $z < 0$ or $z > 0$. Both parts experience two different energy processes, energy charge ($z < 0$) or energy discharge ($z > 0$), which prevents the light beam from diverging in free space. In principle, the non-diffractive distance can be magnified by increasing the strength of the energy oscillation. However, as demonstrated in Supplementary Figures 5 and 6, the energy charge process cannot be strengthened limitlessly because of the finite power in free space. A finite energy charge can support only a finite energy discharge, thus leading to the impossibility of creating a light beam with UAD distance for one particular energy oscillation in free space. Although this power barrier in free space is insurmountable, the energy oscillation mechanism still offers a new possibility to generate light beams with UAD distance. Again, an anti-diffracting light beam experiences a finite energy charge when $z < 0$ as well as a finite energy discharge when $z > 0$. When the light beam completely discharges its energy ($z > 0$), it cannot be recharged again. Thus, the diffraction effect finally dominates, and the light beam can no longer propagate. For this reason, the solution to this problem mainly depends on the ability of the light beam to recharge after the energy is completely discharged.

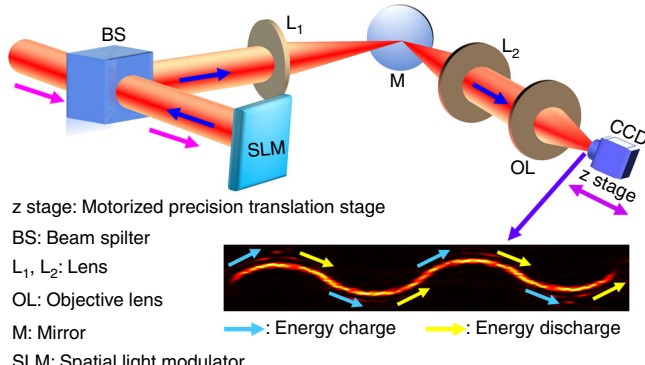

z stage: Motorized precision translation stage
BS: Beam spilter
L₁, L₂: Lens
OL: Objective lens
M: Mirror
SLM: Spatial light modulator

→: Energy charge   →: Energy discharge

**Fig. 1** Schematic of the experimental setup. A collimated incident $x$ linearly polarized Gaussian beam with a wavelength of $\lambda = 632.8$ nm propagating along the optical axis passes through a phase-only SLM (spatial light modulator), two lenses (L₁, L₂) before it is focused by the objective lens (OL) (NA = 0.095). The magenta and blue arrows indicate the propagation directions of incident and modulated beams, respectively. BS and M are beam splitter and mirror. L₁ ($f_1 = 120$ mm) and L₂ ($f_2 = 150$ mm) compose a 4$f$-system that conjugates the phase coded in the SLM and the entrance plane of the OL. UAD light beams with different numbers of energy oscillations can be reconstructed by recording the light intensity in different $z$ planes using a CCD, which can be moved along the optical axis by a motorized precision translation stage ($z$ stage). For example, a UAD light beam with four energy oscillations propagates in an energy charge–discharge–recharge–discharge manner, with the light blue and yellow arrows denoting energy charge and discharge, respectively. One pair of adjacent light blue and yellow arrows denotes an entire energy oscillation

**Critical condition for energy recharge**. Although energy oscillation is a general property shared by all anti-diffracting light beams, not all those light beams are suitable for energy recharge when finishing energy discharge in free space. Energy recharge occurs only at the switch point between adjacent energy oscillations, where one energy oscillation is almost finished performing energy discharge while another is just beginning to recharge. Thus, the energy flux in the former energy oscillation must be the same as that of the latter one at the switch point. Otherwise, the incontinuity of the energy flux will cause interference between adjacent energy oscillations.

Suppose that there are two modes of light beams, and the energy fluxes of which can be expressed as

$$\langle \mathbf{S}_{\delta 1} \rangle_{tP_2} = -\langle \mathbf{S}_{\delta 1} \rangle_{tP_1} \tag{2}$$

$$\langle \mathbf{S}_{\delta 2} \rangle_{tP_2} = -\langle \mathbf{S}_{\delta 2} \rangle_{tP_1}. \tag{3}$$

When the first mode ($\delta 1$) of light beam almost finishes energy discharge $\langle \mathbf{S}_{\delta 1} \rangle_{tP_1}$, the second mode ($\delta 2$) must recharge the energy again so that the light beam can remain anti-diffracting. As a result,

$$\langle \mathbf{S}_{\delta 1} \rangle_{tP_1} = \langle \mathbf{S}_{\delta 2} \rangle_{tP_2}. \tag{4}$$

From Eqs. (2), (3), and (4), the critical condition for energy recharge in free space can be easily obtained, and it can be simplified as

$$\langle \mathbf{S}_{\delta 1} \rangle_{tP_2} = -\langle \mathbf{S}_{\delta 1} \rangle_{tP_1} = -\langle \mathbf{S}_{\delta 2} \rangle_{tP_2} = \langle \mathbf{S}_{\delta 2} \rangle_{tP_1}. \tag{5}$$

That is, only the light beams satisfied in Eq. (5) can be utilized for energy recharge again in free space. As demonstrated in Supplementary Note 2, this critical condition can be easily satisfied by quasi-Airy beams with $\pm\eta$, which can be expressed as (see Supplementary Equation 11)

$$\langle \mathbf{S}_{\eta} \rangle_{tP_2} = -\langle \mathbf{S}_{\eta} \rangle_{tP_1} = -\langle \mathbf{S}_{-\eta} \rangle_{tP_2} = \langle \mathbf{S}_{-\eta} \rangle_{tP_1}. \tag{6}$$

Thus, a quasi-Airy beam with $\eta$ can be recharged by its mutually complementary mode $-\eta$.

**UAD light beam with multiple energy oscillations**. Here, we introduce the multiple energy oscillation mechanism to recharge the energy of a quasi-Airy beam after energy discharge, thus permitting the generation of UAD light beams in free space. As its name implies, this energy mechanism requires multiple energy oscillations in the focal region of the lens, as shown in Fig. 1. For the $j$-th energy oscillation, the cubic phase plate can be expressed as (see Supplementary Equation 9)

$$T_{cj} = \exp\left\{ i\eta_j \frac{k \sin^{\sigma_j}\theta}{\sin^{\sigma_j}\alpha} \left[ \sin^3(\varphi + \varphi_0) + \cos^3(\varphi + \varphi_0) \right] \right\} \tag{7}$$

where $\eta_j$ denotes the period of the cubic phase plate, and $\sigma_j$ is the parameter that controls the distribution of the cubic phase plate, with $\sigma_j = 3$ in this paper; $T_{cj}$ stands for the standard cubic phase; $\phi_0$ is the angle by which whole phase can be rotated; and $\theta$ and $\varphi$ are the convergence angle and the azimuthal angle, respectively. In addition, $\alpha = \arcsin(\text{NA}/n)$, where NA is the numerical aperture of the objective lens, and $n$ is the refractive index in free space.

Since energy recharge occurs only at the switch point between adjacent energy oscillations where the energy flux of the energy discharge process in the initial energy oscillation is equal to that of the energy charge in the second energy oscillation, the parabola-like energy oscillations, which descend when $\eta > 0$ and ascend when $-\eta$, must come in pairs so that the energy of $\langle \mathbf{S}_{\eta} \rangle_{tP_2}$ can be recharged by $\langle \mathbf{S}_{-\eta} \rangle_{tP_1}$ and $\langle \mathbf{S}_{-\eta} \rangle_{tP_2}$ can be recharged by $\langle \mathbf{S}_{\eta} \rangle_{tP_1}$. That is, energy recharge can be realized by simply overlapping $\langle \mathbf{S}_{\eta} \rangle_{tP_2}$, $\langle \mathbf{S}_{-\eta} \rangle_{tP_2}$ with their corresponding $\langle \mathbf{S}_{-\eta} \rangle_{tP_1}$, $\langle \mathbf{S}_{\eta} \rangle_{tP_1}$ at the switch point.

This process requires a highly precise manipulation of each energy oscillation in the focal region of the lens. An optical pen is therefore developed to manipulate the position, number, amplitude, and phase of each energy oscillation simultaneously and precisely so that energy recharge can occur at the switch point and multiple energy oscillations can be realized in free space. According to Supplementary Equation 19, the phase of optical pen can be written as

$$\psi_p = \text{Phase}\left[ \sum_{j=1}^{N} \text{PF}(s_j, x_j, y_j, z_j, \delta_j) \right] \tag{8}$$

where $N$ indicates the number of foci; $x_j$, $y_j$, and $z_j$ denote the position of the $j$-th focus in the focal region; and $s_j$ and $\delta_j$ are weight factors that can be used to adjust the amplitude and phase of the $j$-th focus, respectively.

By combining the optical pen in Eq. (8) with the cubic phase plate in Eq. (7), the final phase for the generation of a UAD light

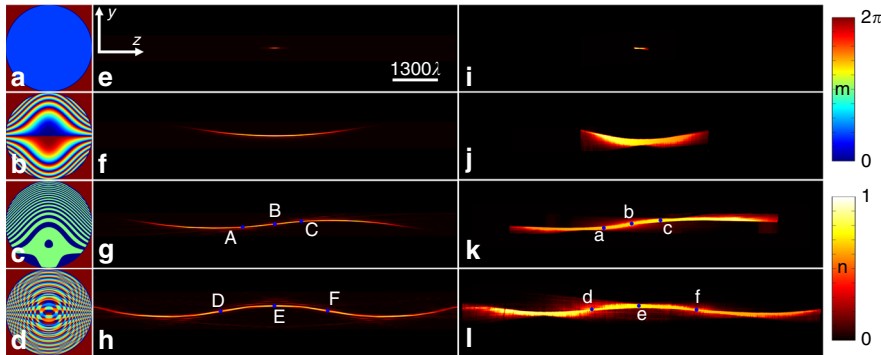

**Fig. 2** UAD light beams with different number of energy oscillations under the condition of NA = 0.095. **e** Zero oscillations, **f** one oscillation, **g** two oscillations, and **h** three oscillations are generated by the phases (**a–d**), the parameters of which can be found in Table 1. The anti-diffracting distances in the experiment are approximately 204.1λ, 2544.9λ, 5606.2λ, and 8316.1λ (**i–l**), and their corresponding theoretical results (**e–h**) are 195.7λ, 2560λ, 5592λ, and 8310λ, respectively. A–F are the points on the UAD light beams, wherein B, D, F are the switch points. a–f are their corresponding points in the experiment. All light intensities of UAD light beams are normalized to a unit value, which are indicated by the color bar (**m**). The color bar (**n**) shows the phase scale of **a–d**

**Table 1 UAD light beams under the condition of NA = 0.095**

| N | $y_1$ | $y_2$ | $y_3$ | $z_1$ | $z_2$ | $z_3$ |
|---|---|---|---|---|---|---|
| 1 | 0 | | | 0 | | |
| 1 | 0 | | | 0 | | |
| 2 | −42.5 | 42.5 | | −1535 | 1535 | |
| 3 | −42.5 | 42 | −42.5 | −3046 | 0 | 3046 |
| $\eta_1$ | $\eta_2$ | $\eta_3$ | $\delta_1$ | $\delta_2$ | $\delta_3$ | Parameter |
| 0 | | | | | | NA = 0.095 |
| −5 | | | | | | $\varphi_0 = 0.75\pi$ |
| −5 | 5 | | −0.07π | 0.07π | | $x_j = 0$ |
| −5 | 5 | −5 | 0.1π | 0 | −0.1π | $s_j = 1$ |

beam with multiple energy oscillations can be simplified to

$$\psi_{UAD} = \text{Phase}\left\{\sum_{j=1}^{N}\left[T_{cj}\text{PF}(s_j, x_j, y_j, z_j, \delta_j)\right]\right\} \quad (9)$$

In this equation, the optical pen is responsible for controlling the number, position, amplitude, and phase of the energy oscillations, while the cubic phase determines the orientation of the energy oscillations. This phase can be considered the Fourier spectrum of the UAD light beam, which can be easily implemented using the phase-only spatial light modulator (SLM) shown in Fig. 1.

**Experiment**. Following the general focusing theory of linearly polarized Gaussian beams presented in Supplementary Equation 3, a UAD light beam with multiple energy oscillations can be obtained by substituting $T_c$ for $T_{UAD} = \exp(i\psi_{UAD})$. Note that energy oscillations occur naturally in free space; thus, a UAD light beam can always be realized with arbitrary NA. Without a loss of generality, we take NA = 0.095 as an example to generate UAD light beams with different numbers of energy oscillations in air, namely, $n = 1$. The corresponding results with NA = 0.8 can be found in Supplementary Figure 13. In the following simulations and experiments, the unit of length in all figures is the wavelength λ, and the light intensity is normalized to the unit value so that the propagation distance of light beam can be manifested clearly. The anti-diffracting distance of the UAD light beam is evaluated with the FWHM (full width at half maximum).

Figure 1 presents the schematic of the experimental setup. A collimated incident x linearly polarized Gaussian beam (wavelength: 632.8 nm) propagating along the optical axis passes through a phase-only SLM and two lenses ($L_1$, $L_2$) before it is focused by the objective lens (OL), with NA = 0.095. $L_1$ ($f_1 = 120$ mm) and $L_2$ ($f_2 = 150$ mm) compose a 4f-system that makes the phase coded in the SLM and the entrance plane of the OL conjugate. The light intensity of the UAD light beam in the x–y plane is recorded using a CCD, which can be moved along the optical axis by a motorized precision translation stage (z stage). By recording the light intensities in different z planes using the CCD, UAD light beams with different numbers of energy oscillations can be reconstructed. The light blue and yellow arrows represent the energy charge and discharge processes, respectively. One pair of adjacent light blue and yellow arrows indicates an entire energy oscillation.

Figure 2i–l, e–h present the experimental and theoretical results of UAD light beams with 0, 1, 2, and 3 energy oscillations, respectively. The phases coded in the SLM are shown in Fig. 2a–d; and the parameters can be found in Table 1. The anti-diffracting distances of the UAD light beams with 0, 1, 2, and 3 energy oscillations in the experiment (Fig. 2i–l) are approximately 204.1λ, 2544.9λ, 5606.2λ, and 8316.1λ, and the corresponding theoretical results in Fig. 2e–h are 195.7λ, 2560λ, 5592λ, and 8310λ, respectively. The deviation ratio between both results is defined as $|\text{ADD}_{ex} - \text{ADD}_{th}|/\text{ADD}_{th}$, where $\text{ADD}_{ex}$ and $\text{ADD}_{th}$ are the anti-diffracting distance for the experimental and theoretical UAD light beams, respectively. The maximum deviation ratio of the UAD light beams in Fig. 2 is 4.29%, and the experimental results are consistent with the theoretical predictions. Generally, a light beam with a longer anti-diffracting distance than that of one energy oscillation has long been considered impossible due to the finite power in free space. However, using the multiple energy oscillation mechanism, UAD light beams can be realized in an energy charge–discharge–recharge–discharge manner, and the anti-diffracting distance of these beams can be extensively promoted by increasing the number of energy oscillations via the optical pen.

To better understand the multiple energy oscillation mechanism, we take the UAD light beam shown in Fig. 2g as an example and investigate the light intensities and transverse energy fluxes (green arrows) in the $z = -1100\lambda$, 0, 1100λ planes, which are denoted by points A, B, and C, respectively. The corresponding light intensities are shown in Fig. 3a–c, whereas the energy fluxes are indicated by the green arrows in Fig. 3m–o. As shown in Fig. 3m–o, Point A experiences an energy discharge in the initial energy oscillation. Point B is the switch point at which the initial energy is nearly fully discharged and the second energy charge is just beginning. Point C

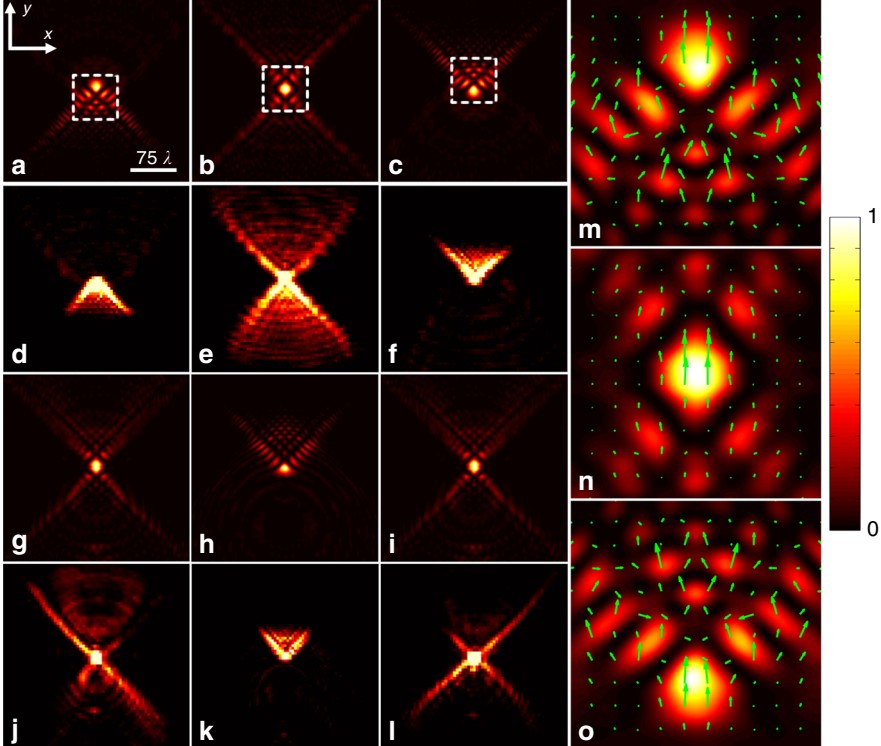

**Fig. 3** Light intensities of UAD light beams in different $z$ planes. **a–c**, **g–i** and **d–f**, **j–l** are the theoretical and experimental results, respectively, where **a**, **d** $z = -1100\lambda$; **b**, **e** $z = 0$; **c**, **f** $z = 1100\lambda$; **g**, **j** $z = -1465\lambda$, **h**, **k** $z = 0$, and **i**, **l** $z = 1465\lambda$. All are denoted by the points a-f and A-F in Fig. 2. Points B, D, F and b, d, f are the theoretical and experimental switch points, respectively. **m-o** are the energy fluxes of **a–c**, which are indicated by the green arrows. All light intensities are normalized to a unit value, which are indicated by the color bar

falls totally within the second energy charge process. If power compensation does not occur at point B, then the light beam may endure significant divergence and will not be able to propagate further in free space, which is similar to the common quasi-Airy beam in Fig. 2f.

However, the UAD light beam is able to recharge again at point B via the multiple energy oscillation mechanism. Therefore, the sidelobes at point B must be cross-shaped so that the bottom and upper energy fluxes are in the same direction, as shown in Fig. 3n. Here, the bottom sidelobe corresponds to an energy discharge process similar to that found at point A in Fig. 3m in the initial energy oscillation, whereas the upper sidelobe corresponds to the subsequent energy charge similar to that at point C in Fig. 3o in the second energy oscillation. During propagation in free space, the initial energy oscillation reduces as the light beam passes through point B and eventually vanishes. As shown in Fig. 3o, the second energy oscillation finally takes over, and the light beam undergoes a second energy charge at point C. In this way, a UAD light beam can propagate over a super-long range by only repeating energy oscillations, which to a large extent circumvents the limitation of the finite power in free space. Their corresponding experimental light intensities are shown in Fig. 3d–f. The same energy process also occurs for the UAD light beam with 3 energy oscillations. Points d and f in Fig. 2l are two switch points in which the energy discharge process is almost finished and a new energy charge process is just beginning. Point e in Fig. 3k remains anti-diffracting in the second energy oscillation process, as predicted in Fig. 3h. Compared with the theoretical results in Fig. 3g,i, the cross-shaped sidelobes in Fig. 3j, l are slightly asymmetric, which is attributed to the deviation of the light path in the experiment.

## Discussion

Creating a UAD light beam in free space involves four key points: revealing an underlying mechanism of anti-diffracting light beams, renewing our understanding of the propagation behavior of anti-diffracting light beams, finding a suitable anti-diffracting light beam for energy recharge, and developing an optical pen for the realization of UAD light beams with multiple energy oscillations. For the first two points, diffraction-free is a general property shared by all anti-diffracting light beams that was previously explained by their mathematical forms derived from the Helmholtz equation[1,3,4,6]. However, by exploring the energy flux of those light beams, energy oscillation is found to represent the physical mechanisms underlying this common property, which confines the energy of anti-diffracting light beams into an interaction between mainlobe and sidelobes so that they will not diverge freely as observed with Gaussian beams during propagation in free space. This mechanism not only renews our understanding of the propagation behavior of anti-diffracting light beams but also provides a possibility of counteracting the diffraction effect in free space. That is, when anti-diffracting light beams finish energy discharge, they cannot be recharged again. This conceptual change paves the way for creating UAD light beams with the multiple energy oscillation mechanism in free space.

Despite the general energy oscillation mechanism, the third point should not be taken to mean that all those light beams are suitable for recharging energy again in free space. One critical condition must be satisfied: the energy fluxes of the initial energy discharge must be equal to those of the secondary energy charge at the switch point between adjacent energy oscillations. Thus, only the light beams satisfied in Eq. (5) can be recharged in free

space. For example, the Bessel beam at point $P_1$ cannot be recharged by the energy flux of point $P_2$ using the optical pen due to their inverse energy fluxes (see Supplementary Figure 7). Overlapping the energy flux of point $P_1$ with that of point $P_2$ will cause an incontinuity of energy fluxes at the switch point, thereby leading to interference at the switch point. Thus, it is impossible to create UAD light beams with the Bessel beam as their base. In contrast, a quasi-Airy beam possesses a pair of mutually complementary modes with $\pm\eta$. Equation (6) shows that a quasi-Airy beam of $\eta$ can be recharged simply by using that of $-\eta$. Hence, a UAD light beam can propagate much farther than a Bessel beam[1,2] and quasi-Airy beam[6–9] simply by repeating energy oscillations in free space.

For the fourth point, the optical pen is another crucial aspect responsible for the realization of the multiple energy oscillation mechanism. As a versatile optical tool, the optical pen has an explicit form (Eq. (8)) that can be used to unify the relationship between the focal pattern and the phase in the entrance plane. Taking the focus array in Supplementary Figure 10 as an example, the optical pen represents all possible phases using only two different weight factors, $s_j$ and $\delta_j$. Thus, the optical pen can flexibly adjust the position, number, amplitude and phase of each focus in the focal region simultaneously and precisely, which ensures that energy recharge can occur at the switch points between adjacent energy oscillations. This advantage makes the optical pen the perfect optical tool for the creation of UAD light beams.

When finishing the above processes, UAD light beams can be built up by concatenating a sequence of energy oscillations with alternate signs of their curvatures. However, the UAD light beam is a new type of light beam in free space and not just multiple foci in the focal region. First, as a light beam, the energy flux must evolve continually so that the information is not lost during propagation. Since Eq. (5) is satisfied, a quasi-Airy beam can be recharged at the switch point with its mutually complementary mode. Therefore, the energy flux of UAD light beams can evolve without mutation during propagation in free space. Second, the UAD light beams generated here can be considered a sum of Airy beam series with different weight factors, which can be simplified as

$$\mathrm{UAD} = \sum_{j=1}^{N} s_j AiB_j(x - \Delta x_j, y - \Delta y_j, z - \Delta z_j) \qquad (10)$$

where $AiB_j(x-\Delta x_j,\ y-\Delta y_j,\ z-\Delta z_j)$ represents the $j$-th energy oscillation with the position $(\Delta x_j, \Delta y_j, \Delta z_j)$ and the weight factor $s_j = 1$, which can be flexibly adjusted by the optical pen. As shown in Eq. (10), the UAD light beam is a special solution of the Helmholtz equation. Third, the UAD light beam propagates in a wavy trajectory, while the Airy beam can propagate only parabolically. That is, the UAD light beam is not a type of Airy beam. Based on the above reasons, one can conclude that the UAD light beam is an entirely new anti-diffracting light beam in free space.

To some extent, UAD light beams are similar to light beams generated in nonlinear materials[24–29] such as spatial light solitons. Both types of light beams can propagate without diverging by suppressing the diffraction effect. However, the method of counteracting the diffraction effect is totally different. Without taking advantage of light-material interactions, the creation of the UAD light beam in free space conveys a physical idea that the diffraction effect can be suppressed by the property of the anti-diffracting light beam itself, namely, energy oscillation. Energy oscillation is a directional energy flux shared by all anti-diffracting light beams that confines the energy into an interaction between the mainlobe and sidelobes. Thus, if only energy

oscillations occur, those light beams can preserve their shapes without divergence during propagation in free space. Accordingly, UAD light beams can be considered a similar spatial optical soliton of free space.

In conclusion, we have theoretically and experimentally demonstrated that light beams with UAD distances can be generated via the multiple energy oscillation mechanism in free space. The non-diffractive distance can be manipulated by on-demand tuning the number of energy oscillations using the optical pen. For example, UAD light beams with 3 energy oscillations can propagate without significant divergence over $8316.1\lambda$ when NA = 0.095, whereas a quasi-Airy beam can only propagate over $2544.9\lambda$. Due to their super-long non-diffractive distances, UAD light beams offer promising applications in many scientific studies, such as optical imaging[14–16], laser-assisted guiding of electric discharges[23], optical trapping[17–19], and optical communication[20–22].

In addition to the potential practical applications, this work may have a considerable impact on the optics community as well as other research areas, such as the study of electrons and acoustics. First, although the energy oscillation mechanism satisfying Eq. (5) was first found for a quasi-Airy beam, other accelerating light beams, such as the Mathieu and Weber beam[3,4], may also possess a similar mechanism. Thus, other types of UAD light beams with the multiple energy oscillation mechanism can also be created in free space, which may spark a new era regarding the study of similar spatial light solitons in free space. Second, since electron Airy beams and self-bending acoustic beams have already been demonstrated in free space[33,34], this work may inspire the creation of their counterpart UAD light beams and solve certain problems in both fields.

## Data availability

All data supporting the findings of this study are available from the corresponding author on request.

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

## Acknowledgements

Parts of this work were supported by the National Basic Research Program of China (2015CB352001); National Natural Science Foundation of China (61525503/61620106016/11804232/61835009); the Guangdong Natural Science Foundation Innovation Team (2014A030312008); the Hong Kong, Macao and Taiwan Cooperation Innovation Platform & major projects of international cooperation in Colleges and Universities in Guangdong Province (2015KGJHZ002); the Shenzhen Basic Research Project (JCYJ20150930104948169/JCYJ20160328144746940/GJHZ20160226202139185); and the National Key Research and Development Program of China (2016YFF0101603).

## Author contributions

X. Weng conceived of the research and designed the experiments. Q. Song and X. Weng performed the experiments and analyzed all of the data. X. Li, X. Gao and H. Guo supervised the experiments. X. Weng wrote the paper, and H. Guo offered advice regarding its development. J. Qu and S. Zhuang directed the entire project.

## Additional information

**Competing interests:** The authors declare no competing interests.

