## [Peer Review File · Nature Communications]

Reviewers' comments:

Reviewer #1 (Remarks to the Author):

The authors have addressed all my concerns in a very convincing manner. As such I support publication of this interesting article in Nature Communications.

Reviewer #2 (Remarks to the Author):

Manuscript Title: Free-space creation of ultralong anti-diffracting light beam with 2 multiple energy oscillations adjusted using 'optical pen'

Authors report on a quite original method to create an ultralong antidiffracting (UAD) beam, as they call it, in the linear regime of optics, i.e., avoiding nonlinear effects. Results are very nice but I have one main and big concern about the physical claim, the clarification of which is crucial before the article could be published. I elaborate on this concern below.

The sentence "The anti-diffracting distance is no longer restricted by finite power in free space but instead depends on the number of energy oscillations" already in lines 18-19 is profoundly misleading and rises strong doubts about a complete "new understanding of non-diffractive light beams", as stated on line 20. To the referee's understanding, what authors are doing in this work is to superimpose in a single phase-mask various Airy beams with their intensity maxima placed at different locations in space. Therefore the input laser light power is divided by 'N' where 'N' is the number of the Airy beams written in the phase mask. Therefore, each Airy beam will 'dispose' of $1/N$ of the input light energy. How can this then be claimed to be not limited by the input power? In this way of concatenating N-Airy beams, each $1/N$ -th portion of the energy of light, once focused around the most intense part of the corresponding Airy beam then diffracts irreversibly, and the apparent 'non-diffracting' behaviour simply comes from the fact that the next $1/N$ -th portion of the light beam is focusing just after it. Even if this is a very ingenious way of concatenating light beams, the effect is intrinsically limited by diffraction, conversely to what authors state.

Further examples in text where the same claim is expressed in various different ways:

Lines 59-61: "The non diffractive distance of a UAD light beam is determined only by the number of energy oscillations rather than the finite power in free space."

Lines 106-107: "(...) the energy oscillation mechanism still offers a new possibility of transforming this power-dependent distance into a power-independent one."

Lines 110-112: "For this reason, the solution to this problem is no longer restricted by the finite power in free space but instead depends on the ability of the light beam to recharge after the energy is completely discharged. "

Lines 220-221: "In this way, a UAD light beam can propagate over a super-long range by only repeating energy oscillations without being restricted by the finite power in free space".

Results of this paper are very nice and have their own merit, originality, and beauty. However the main claim is that this effect is not limited by input power, which seems to violate energy conservation itself, and must be clearly presented with its true limitations. The message of the paper should then adjust to the realistic scenario. If the referee is wrong with the above interpretation, then it should be explicitly explained why their approach is not 'limited by input power nor diffraction'. While this crucial point is not clarified the referee cannot recommend acceptance of the manuscript. The only way to clearly demonstrate that such effect is not limited

by power is to make a figure similar to Fig. 2(l) with, say, six energy oscillations rather than three, keep input laser power constant and obtain the same peak intensity as in the current Fig. 2(l). The guess of the referee, based on energy conservation arguments is that peak intensity will decrease by roughly a factor of 2 (when increasing the number of energy oscillations from 3 to 6).

For Reviewer#1

Reviewer #1 (Remarks to the Author):

The authors have addressed all my concerns in a very convincing manner. As such I support publication of this interesting article in Nature Communications.

Answer: We are grateful for the reviewer's efforts to improve the quality of our paper.

For Reviewer#2

Reviewer #2 (Remarks to the Author):

Manuscript Title: Free-space creation of ultralong anti-diffracting light beam with 2 multiple energy oscillations adjusted using ‘optical pen’. Authors report on a quite original method to create an ultralong anti-diffracting (UAD) beam, as they call it, in the linear regime of optics, i.e., avoiding nonlinear effects. Results are very nice but I have one main and big concern about the physical claim, the clarification of which is crucial before the article could be published. I elaborate on this concern below.

The sentence “The anti-diffracting distance is no longer restricted by finite power in free space but instead depends on the number of energy oscillations” already in lines 18-19 is profoundly misleading and rises strong doubts about a complete “new understanding of non-diffractive light beams”, as stated on line 20. To the referee’s understanding, what authors are doing in this work is to superimpose in a single phase-mask various Airy beams with their intensity maxima placed at different locations in space. Therefore the input laser light power is divided by ‘N’ where ‘N’ is the number of the Airy beams written in the phase mask. Therefore, each Airy beam will ‘dispose’ of $1/N$ of the input light energy. How can this then be claimed to be not limited by the input power? In this way of concatenating N-Airy beams, each $1/N$ -th portion of the energy of light, once focused around the most intense part of the corresponding Airy beam then diffracts irreversibly, and the apparent ‘non-diffracting’ behavior simply comes from the fact that the next $1/N$ -th portion of the light beam is focusing just after it. Even if this is a very ingenious way of concatenating light beams, the effect is intrinsically limited by diffraction, conversely to what authors state.

Further examples in text where the same claim is expressed in various different ways:

Lines 59-61: “The non diffractive distance of a UAD light beam is determined only by the number of energy oscillations rather than the finite power in free space.”

Lines 106-107: “(…) the energy oscillation mechanism still offers a new 107 possibility of transforming this power-dependent distance into a power-independent one.”

Lines 110-112: “For this reason, the solution to this problem is no longer restricted by the finite power in free space but instead depends on the ability of the light beam to recharge after the energy is completely discharged. ”

Lines 220-221: “In this way, a UAD light beam can propagate over a super-long range by only repeating energy oscillations without being restricted by the finite power in free space”.

Results of this paper are very nice and have their own merit, originality, and beauty. However the main claim is that this effect is not limited by input power, which seems to violate energy conservation itself, and must be clearly presented with its true limitations. The message of the paper should then adjust to the realistic scenario. If the referee is wrong with the above interpretation, then it should be explicitly explained why their approach is not ‘limited by input power nor diffraction’. While this crucial point is not clarified the referee cannot recommend acceptance of the manuscript. The only way to clearly demonstrate that such effect is not limited by power is to make a figure similar to Fig. 2(l) with, say, six energy oscillations rather than three, keep input laser power constant and obtain the same peak intensity as in the current Fig. 2(l). The guess of the referee, based on energy conservation arguments is that peak intensity will decrease by roughly a factor of 2 (when increasing the number of energy oscillations from 3 to 6).

Response to the reviewer's concerns

1. **Question:** The sentence “The anti-diffracting distance is no longer restricted by finite power in free space but instead depends on the number of energy oscillations” already in lines 18-19 is profoundly misleading and rises strong doubts about a complete “new understanding of non-diffractive light beams”, as stated on line 20. To the referee’s understanding, what authors are doing in this work is to superimpose in a single phase-mask various Airy beams with their intensity maxima placed at different locations in space. Therefore, the input laser light power is divided by ‘N’ where ‘N’ is the number of the Airy beams written in the phase mask. Therefore, each Airy beam will ‘dispose’ of 1/N of the input light energy. How can this then be claimed to be not limited by the input power?

Answer: Thank you very much for the professional comments! We are really sorry for the misunderstanding about the definition difference between the ‘Input power’ and ‘Finite power’ caused by our unclear statements in the paper. As shown in Fig. 1, *the ‘Input power’ mentioned by the reviewer is the energy of incident light beam before the objective lens, while the ‘Finite power’ is the energy of anti-diffracting light beam itself, which can be obtained by the integration of the amplitude squared along the propagation trajectory of anti-diffracting light beam.*

Fig. 1 The schematic of focusing system.

To address this problem, we are going to explain the difference between the ‘Input power’ and the ‘power’ of anti-diffracting light beams in details. For the ‘Input power’, energy oscillations with larger number would weaken the light intensity of the N-th energy oscillation in UAD light beam. For instance of the UAD light beams with 1, 2, 3 energy oscillations in Fig. 2, we suppose the ‘Input power’ of all these beams as $P=1$. Without consideration of absorption, the light intensity of the light beam with one energy oscillation can be defined as $P=1$, whereas the light intensities of UAD light beams with 2, 3 energy oscillations are accordingly divided by 2, 3, as shown in Fig.2(a-c). Hence, *the anti-diffracting distance of light beam is inherently irrelevant with the ‘Input power’*. Without using Multiple energy oscillation mechanism, only one energy oscillation of light beam can be generated in free space, as shown in Fig. 2(a). **Larger ‘Input power’ can only lead to a brighter light beam, but has no impact on the anti-diffracting distance.** Thus, all light intensities of UAD light beams are being normalized in the paper so that the brightness of UAD light beams remain the same, as seen in Fig. 2(a₁-c₁).

In our paper, we adopt the concept of ‘Finite power’ as an intrinsic attribute of the anti-diffracting light beams. Note that the ‘Finite power’ is prevalent in the regime of anti-diffracting light beams: see the original and fundamental references [1-7]. To be consistent with these references, *the ‘Finite power’ mentioned in the paper is defined as the integration of the amplitude square along the propagation trajectory of anti-diffracting light beams.* For an ideal anti-diffracting light beam with infinite anti-diffracting distance, the ‘Infinite power’ is needed to maintain its shape; for a quasi-anti-diffracting light beam, it only possesses the ‘Finite power’, resulting in a finite anti-diffracting distance. *By contrast with the ‘Input power’, the ‘Finite power’ is only relevant with the anti-diffracting distance, rather than the brightness.* Thus, we normalize the light intensities of UAD light beams to a unit value mathematically so that the anti-diffracting distance can be easily manifested. *It should be emphasized that the absolute ‘Power’ still relates to the ‘Input power’ and obeys the energy conservation.*

Fig.2 The light intensity of UAD light beam with 1, 2, 3 energy oscillations, where (a-c) Non-normalized light intensity; (a₁-c₁) the corresponding normalized light intensity.

The understanding way of the ‘Finite power’ of anti-diffracting light beam can also be interpreted theoretically in the mathematical form. We take the ideal Bessel and Airy beam as example. As shown in Fig. 3(a), an ideal Bessel beam with an amplitude proportional to $J_n(k_r r) \exp(-ik_z z)$ can preserve its shape infinitely without divergence during propagation in free space. Thus, the ‘power’ of the ideal Bessel beam can be obtained by [1]

$$I_B = \left| \int \int_{-\infty}^{+\infty} J_n(k_r r) \exp(-ik_z z) dr dz \right|^2 \rightarrow \infty. \quad (1)$$

Similarly, the ‘power’ of the ideal Airy beam in Fig. 3(b) can be obtained by [2]

$$I_A = \left| \int \int_{-\infty}^{+\infty} Ai(s - \xi^2 / 4) \exp[i(s\xi / 2 - \xi^3 / 12)] dx dz \right|^2 \rightarrow \infty. \quad (2)$$

where $s = x / x_0$, $\xi = z / kx_0^2$, and x_0 is an arbitrary constant. Since the Bessel and Airy function in Eq. (1,2) are not square integrable, both light beams possess ‘Infinite power’ in free space. However, such ‘Infinite power’ can only be achieved by the infinite aperture of the lens. *In practice, finite apertures can only transfer ‘Finite power’ to an anti-diffracting light beam, thereby leading to a finite anti-diffracting distance in free space.* For this reason, the creation of

UAD light beams are always considered to be impossible in free space. In our paper, we intend to overcome the limitation of the ‘Finite power’ by means of multiple energy oscillation mechanism. Although the ‘Finite power’ in free space is intrinsically associated with the integral path, we are able to significantly elongate the anti-diffracting distance by on-demand tuning the number of energy oscillations, which to a large extent circumvents the limitation of the finite power in free space.

Fig.3 The light intensity of ideal Bessel beam (a) and Airy beam (b)

To make it more understandable, we revise the paper according to the reviewer’s comments. Firstly, we give a clear definition of ‘Infinite power’ for an Anti-Diffracting Light Beam in the **Supplementary information: Section 1 (highlighted by red color)**. Secondly, we revise the sentence ‘In the following simulations and experiments, the unit of length in all figures is the wavelength λ , and the light intensity is normalized to the unit value’ to ‘In the following simulations and experiments, the unit of length in all figures is the wavelength λ , and the light intensity is normalized to the unit value so that the propagation distance of light beam can be manifested clearly.’ [See Main text: **Section Experiment: First paragraph, highlighted by red color**].

2. **Question:** In this way of concatenating N-Airy beams, each 1/N-th portion of the energy of light, once focused around the most intense part of the corresponding Airy beam then diffracts irreversibly, and the apparent ‘non-diffracting’ behavior simply comes from the fact that the next 1/N-th portion of the light beam is focusing just after it. Even if this is a very ingenious way of concatenating light beams, the effect is intrinsically limited by diffraction, conversely to what authors state.

Answer: We agree with the referee’s insight comment on that the N-th portion of the UAD light beam is actually limited by diffraction at the discharge part. In our paper, UAD light beam can be understood from the perspective of concatenating N-Airy beams. However, this process requires a high-precision and synergistic manipulation of each energy oscillation, such as the position, number, amplitude and phase, so that the shape and light intensity of UAD light beam can be preserved during propagation. For one particular energy oscillation within the UAD light beam, the diffraction effect would finally dominate when it completely discharges its energy. Thus, it is undoubtedly that the N-th energy oscillation is limited by the diffraction. *However, the non-diffracting behavior is an overall phenomenon, not a partial one. Namely, the UAD light beam with multiple energy oscillations is a new type of anti-diffracting light beam in free space, not just multiple Airy beams in the focal region.* We explicitly explain the reasons as follows:

- A. As a light beam, the energy flux must evolve continually so that the information is not lost during propagation. Since Eq. (3) is satisfied, a quasi-Airy beam can be recharged at the switch point with its mutually complementary mode. Therefore, the energy flux of UAD light beams can evolve without mutation during propagation in free space.

$$\langle \mathbf{S}_{\delta 1} \rangle_{r_2} = -\langle \mathbf{S}_{\delta 1} \rangle_{r_1} = -\langle \mathbf{S}_{\delta 2} \rangle_{r_2} = \langle \mathbf{S}_{\delta 2} \rangle_{r_1} \quad . \quad (3)$$

- B. The UAD light beams generated here can be considered a sum of Airy beam series with different weight factors, which can be simplified as

$$UAD = \sum_{i=1}^N s_i AiB_i(x - \Delta x_i, y - \Delta y_i, z - \Delta z_i) \quad (4)$$

where $AiB_i(x - \Delta x_i, y - \Delta y_i, z - \Delta z_i)$ represents the i -th energy oscillation with the position and the weight factor s_i , which can be flexibly adjusted by the optical pen. As shown in Eq. (4), the UAD light beam is a special solution of the Helmholtz equation.

- C. The UAD light beam propagates in a wavy trajectory, while the Airy beam can propagate only parabolically. That is, *the UAD light beam is not a type of Airy beam.*

Based on the above reasons, one can conclude that the UAD light beam is an entirely new anti-diffracting light beam in free space. *Thus, for one particular energy oscillation, the diffractive effect makes it divergence when finishing energy discharge, but for the whole light beam, the UAD light beam can be recharged again with the aid of second energy oscillation. That is, the N -th energy oscillation is limited by the diffraction, but the whole UAD light beam remains anti-diffracting in free space.* In this viewpoint, the whole UAD light beam can propagate invariantly in an energy charge-discharge-recharge-discharge way, which to a large extent avoids the restriction of finite power in free space.’ [See Main text Section: Discussion, Fourth paragraph]

3. **Question:** Results of this paper are very nice and have their own merit, originality, and beauty. However the main claim is that this effect is not limited by input power, which seems to violate energy conservation itself, and must be clearly presented with its true limitations.

Answer: Thank you very much for the comments! In our paper, we intent to generate a UAD light beam that can break through the limitation of ‘Finite power’ in free space. Although we can significantly elongate the anti-diffracting distance by on-demand tuning the number of energy oscillations, we believe that there is a limit because diffraction effect is a nature of light beam. Longer UAD light beams we want to create in free space, stronger diffraction effect it should be overcome. Thus, larger number of energy oscillations are more difficult to create than that of smaller number. As far as we know, UAD light beams with 2, 3 energy oscillations are very unimaginable in free space. If the readers want to create UAD light beam with more number of energy oscillations, more effect should be made. To be honest, UAD light beam with 6 energy oscillations can be created in the focal region without much difficulty. This is the first paper to create UAD light beams in free space. We believe that the readers will get inspirations from it and more works need to be done in the future.

Fig.4 The light intensity of UAD light beams with Four(a), Five (b) and Six (c) energy oscillations.

4. **Question:** In Further examples in text where the same claim is expressed in various different ways: The sentence “The anti-diffracting distance is no longer restricted by finite power in free space but instead depends on the number of energy oscillations” already in lines 18-19.

Lines 59-61: “The non-diffractive distance of a UAD light beam is determined only by the number of energy oscillations rather than the finite power in free space.”

Lines 106-107: “(…) the energy oscillation mechanism still offers a new possibility of transforming this power-dependent distance into a power-independent one.”

Lines 110-112: “For this reason, the solution to this problem is no longer restricted by the finite power in free space but instead depends on the ability of the light beam to recharge after the energy is completely discharged.”

Lines 220-221: “In this way, a UAD light beam can propagate over a super-long range by only repeating energy oscillations without being restricted by the finite power in free space”.

Answer: Thank you very much for the comments! We revised the above strong claims so that the paper can be more readable and acceptable.

- A. We revised lines 18-19 to ‘The anti-diffracting distance can be adjusted by the number of energy oscillations.’
- B. We revised lines 59-61 to ‘Based on the energy oscillation mechanism, the main reason of the finite anti-diffracting distance is that when an anti-diffracting light beam completely discharges its energy, it cannot recharge again. A versatile ‘optical pen’ is therefore developed to manipulate the number, amplitude, position and phase of energy oscillation so that energy recharge can occur in free space and multiple energy oscillations can be realized. Eventually, UAD light beams with a tunable number of energy oscillations can be obtained, which to a large extent avoids the restriction of finite power in free space.’
- C. We revised lines 106-107 to ‘Although this power barrier in free space is insurmountable, the energy oscillation mechanism still offers a new possibility to generate light beams with UAD distance.’
- D. We revised lines 110-112 to ‘For this reason, the solution to this problem mainly depends on the ability of the light beam to recharge after the energy is completely discharged.’
- E. We revised lines 220-221 to ‘In this way, a UAD light beam can propagate over a super-long range by only repeating energy oscillations, which to a large extent circumvents the limitation of the finite power in free space.’

F. In addition, we also revised lines 293-294: ‘The non-diffractive distance is no longer restricted by the finite power in free space but depends on the number of energy oscillations, which can be flexibly manipulated using the ‘optical pen’.’ to ‘The non-diffractive distance can be manipulated by on-demand tuning the number of energy oscillations using the ‘optical pen’.’

Conclusion

Finally, we thank the reviewer for the valuable time spent on our paper so that it can be more readable and understandable. We sincerely hope our answers can address all the reviewer’s concerns.

References

1. J. Durnin, J. J. Miceli, J. H. Eberly, Diffraction-free beams. *Phys. Rev. Lett.* **58**, 1499-1501 (1987).
2. G. A. Siviloglou, D. N. Christodoulides, Accelerating finite energy Airy beams. *Opt. Lett.* **32**, 979-981 (2007).
3. G. A. Siviloglou, J. Broky, A. Dogariu, D. N. Christodoulides, Observation of Accelerating Airy Beams. *Phys. Rev. Lett.* **99**, 213901 (2007).
4. H. Wang, L. Shi, B. Lukyanchuk, C. Sheppard, C. T. Chong, Creation of a needle of longitudinally polarized light in vacuum using binary optics. *Nat. Photonics* **2**, 501-505 (2008).
5. M. Mazilu, D. J. Stevenson, F. Gunn-Moore, K. Dholakia, Light beats the spread: “non-diffracting” beams. *Laser & Photonics Reviews* **4**, 529-547 (2010).
6. P. Zhang, Y. Hu, T. C. Li, D. Cannan, X. B. Yin, R. Morandotti, Z. G. Chen, X. Zhang, Nonparaxial Mathieu and Weber Accelerating Beams. *Phys. Rev. Lett.* **109**, 193901(2012).
7. M. A. Bandres, B. M. Rodriguez-Lara, Nondiffracting accelerating waves: Weber waves and parabolic momentum. *New J Phys* **15**, 013054 (2013).

REVIEWERS' COMMENTS:

Reviewer #2 (Remarks to the Author):

I am glad to see that authors took seriously all my concerns and clarified them unambiguously in the reply and in the modified article. I am therefore in favour of publication of the manuscript in its present form.